# Novel Time-Lapse Parameters Correlate with Embryo Ploidy and Suggest an Improvement in Non-Invasive Embryo Selection

**DOI:** 10.3390/jcm12082983

**Published:** 2023-04-19

**Authors:** Clara Serrano-Novillo, Laia Uroz, Carmen Márquez

**Affiliations:** Gravida, Hospital de Barcelona, 08034 Barcelona, Spain

**Keywords:** embryo quality, preimplantation genetic testing, morphokinetics, time-lapse, ploidy

## Abstract

Selecting the best embryo for transfer is key to success in assisted reproduction. The use of algorithms or artificial intelligence can already predict blastulation or implantation with good results. However, ploidy predictions still rely on invasive techniques. Embryologists are still essential, and improving their evaluation tools can enhance clinical outcomes. This study analyzed 374 blastocysts from preimplantation genetic testing cycles. Embryos were cultured in time-lapse incubators and tested for aneuploidies; images were then studied for morphokinetic parameters. We present a new parameter, “st_2_, start of t_2_”, detected at the beginning of the first cell cleavage, as strongly implicated in ploidy status. We describe specific cytoplasmic movement patterns associated with ploidy status. Aneuploid embryos also present slower developmental rates (t_3_, t_5_, t_SB_, t_B_, cc3, and t_5_-t_2_). Our analysis demonstrates a positive correlation among them for euploid embryos, while aneuploids present non-sequential behaviors. A logistic regression study confirmed the implications of the described parameters, showing a ROC value of 0.69 for ploidy prediction (95% confidence interval (CI), 0.62 to 0.76). Our results show that optimizing the relevant indicators to select the most suitable blastocyst, such as by including st_2_, could reduce the time until the pregnancy of a euploid baby while avoiding invasive and expensive methods.

## 1. Introduction

The selection of the best embryo for transfer is key to success in in vitro fertilization (IVF) treatments. Identifying the embryo with the greatest potential for producing an evolutive pregnancy results in better clinical outcomes while minimizing the associated risks, multiple gestations, and potential complications, both maternal and fetal [1,2]. Embryo evaluation is mainly based on morphological criteria, such as fragmentation, multinucleation or cell size and number, following common grading methods [3]. Major advances in assisted reproductive technologies, especially regarding embryo culture, have allowed for the extension of long-term cultures up to the blastocyst stage, enabling better embryo scoring parameters [4,5,6].

However, while these techniques have been useful for more than 30 years, there is an evident lack of information considering that embryonic development follows a sequence of timed and coordinated events, which require specific developmental rates. Time-lapse (TL) technology solved the problem of static observations and opened new horizons for the study of these dynamic processes. Since the introduction of TL technology, new kinetic markers have been identified and are associated with higher implantation rates. Using this non-invasive scoring method, several studies have found an association between human embryo ploidy and morphokinetics: slower progression, delayed blastulation, and specific cleavage times, such as t3 or the interval t5–t2, have previously been associated with chromosomal aberrations [7,8,9,10].

Nevertheless, clinical pregnancy rates remain at ≈30% [11], a relatively low figure. Clinicians and embryologists urge to find new approaches to improve the selection method and raise these rates without increasing the number of embryos transferred. Genetic screening of embryos may seem a promising solution for this issue, as several studies demonstrate chromosomal alterations to be one of the most common causes of abnormal embryos in IVF (and, thus, of poor clinical outcomes) [12,13,14]. Aneuploid embryos, which often present apparently good development and morphology, are associated with implantation failure, miscarriage, and congenital defects [15].

Preimplantation genetic testing (PGT) for aneuploidies allows for the selection of chromosomally normal embryos. This can raise implantation rates up to 70% when transferring an euploid embryo [16]. However, PGT is not always possible or indicated. Some risks are associated with the technique, as it is an invasive methodology and not completely reliable. The possibility of misdiagnosis, an embryo lacking a diagnosis, or inconclusive results should be considered. Furthermore, the financial cost of embryo testing is elevated; there may be social reasons to discard the embryos, and some clinics do not have the advanced technology required. Therefore, new non-invasive embryo selection methods need to be studied. The search for an association between morphokinetic variables and aneuploidy has recently gained attention, with researchers seeking to find models or algorithms that could help to predict embryo ploidy status, most recently with the help of artificial intelligence (AI). Most authors agree on the importance of early cleavages and also associate delayed blastulation with chromosomal abnormalities, but they apply different pivotal parameters in their predicting models [17,18,19]. In this context, risk models are often not applicable to all TL devices and clinics and depend on the subjectivity or working methodology of each team. The use of AI is promising and can already predict blastulation and implantation with good results, but it cannot yet predict ploidy. Moreover, AI presents some other limitations at present (i.e., the software needs to be trained and often corrected by embryologists, embryos with aberrant development are often miss-annotated, it represents an expensive investment for small clinics, etc.). Therefore, embryologists are still essential for embryo evaluation and the selection of the best embryos for transfer.

The aim of this study is to understand the implications and relation of several classic and novel morphokinetic markers to embryo behavior and to elucidate their ploidy prediction potential, incorporating this information as a key selection tool that is accessible to all. We describe a strategy for predicting the ploidy status of an embryo while avoiding invasive methods and establish the best indicators to select the most suitable blastocyst to achieve a pregnancy with a euploid baby.

## 2. Materials and Methods

### 2.1. Study Design

This retrospective study was conducted at the Gravida Fertilitat Avançada center in Barcelona, Spain. The cohort of the study was drawn from a total of 85 treatments of 73 patients, including 374 blastocysts fertilized using both conventional in vitro fertilization (IVF) (*n* = 202) and intracytoplasmic sperm injection (ICSI) (*n* = 172) in cycles from January 2018 to February 2020. Owing to its retrospective nature and the absence of a priori intervention, the present study did not require the approval of an external review board. The data collection is part of the routine clinical procedure, is not sensitive, and is de-identified; therefore, this observational study posed no risk in terms of compromising the identity or safety of the studied individuals. Among the cases included in the study, with a total of 674 embryos fertilized, 374 (55.5%) of the embryos developed to the blastocyst stage and were able to be biopsied and therefore analyzed by PGT for aneuploidies on the fifth or sixth day of culture (D5 or D6). The annotation of morphokinetic characteristics was performed on all available and analyzed embryos, and the study was blinded to ploidy. Embryos with incomplete annotations failed amplification, and/or abnormal or no fertilization were excluded from the analysis.

This study included patients undergoing PGT due to recurrent miscarriage, repeated implantation failure, advanced maternal age, altered karyotype, and also those doing so electively. A minimum number of embryos to proceed with PGT was not set. The female age ranged from 26 to 45 years (mean ± standard deviation, 38.0 ± 3.9).

### 2.2. Ovarian Stimulation

The ovarian stimulation protocol was based on the patient’s age, hormone levels, antral follicle counts, and prior treatments. Briefly, recombinant alpha or beta FSH (with or without urinary menotropins) were used for ovarian stimulation, ranging from 150 to 375 IU, according to the patient type and ovarian reserve (Gonal F^®^ (Merck, Darmstadt, Germany) or Menopur^®^ (Ferring, Kastrup, Denmark)). To prevent ovulation before egg collection, GnRH antagonists (Orgalutran^®^ (Organon, Amsterdam, The Netherlands) were administered daily starting on day 5 or 6 after FSH administration. Final follicular maturation was triggered with hCG and/or a GnRH agonist (Ovitrelle^®^ (Merck, Darmstadt, Germany) and Decapeptyl^®^ (Ipsen, Boulogne-Billancourt, France), respectively). GnRH agonist was used when patients had ≥10 follicles over 14 mm or estradiol levels ≥2500 pg/mL. If lower levels of follicles were found, a combination of hCG and GnRH was administered.

### 2.3. Oocyte Retrieval and Embryo Culture

Oocytes were retrieved 36 h after the follicular maturation trigger using a transvaginal ultrasound-guided needle. The procedure was performed with the patients under sedation. Oocytes were incubated for around 3 h in Global for Fertilization^®^ medium (LifeGlobal, Guilford, USA) under an oil overlay (Ovoil^TM^, Vitrolife, Gothenburg, Sweden) at 37 °C and 6.6% CO_2_ and 5% O_2_ (previously equilibrated overnight). For ICSI samples, cumulus cells surrounding the oocytes were removed enzymatically with hyaluronidase (HYASE^TM^, Vitrolife) and by pipetting, and ICSI was performed on metaphase II oocytes, which were placed on individual microwells of Geri^®^ (Merck) TL or conventional culture plates, depending on the case. For conventional IVF, oocyte–cumulus complexes were coincubated with sperm overnight. Fertilization was checked 16–18 h post-insemination. Normally fertilized oocytes from IVF were also placed in individual multiwells of the Geri^®^ or conventional culture plates and loaded into the Geri^®^ system or conventional incubator, respectively. Both culture methods were performed in Global Total LP^®^ medium (Vitrolife) covered in mineral oil for up to 6 days at 37 °C with 6–6.7% CO_2_ and 5% O_2_. The culture was briefly interrupted on day 3 in order to perform the assisted hatching and ensure that all embryos were able to hatch prior to the biopsy.

For the TL technology embryos, images of each fertilized oocyte were acquired automatically every 5 min through 11 focal planes. These time-lapse images were used for the assessment of the embryo’s development and to identify the precise timing of developmental events as they were assessed by the embryologists studying the images.

### 2.4. Trophectoderm Biopsy and PGT

Biopsy was performed on day 5/6 of development in all high-quality blastocytes, including grading A and B for the inner cell mass (ICM) and trophoectoderm (TE). Briefly, a combination of laser pulses and mechanical separation was performed to achieve 5–10 TE cells to be used for genetic screening. The biopsied cells were washed and collected in sterile PCR tubes. After the TE biopsy, the embryos were vitrified according to the manufacturer’s instructions (Kitazato, Shizuoka, Japan).

Biopsied cells were diagnosed using whole-genome amplification (NGS, Next Generation Sequencing) by an external genetic analysis laboratory. The NGS platform was used to detect 24-chromosome aneuploidies from a single whole-genome sample. The copy-number values were determined to identify deviations (positive for gains and negative for losses). Each chromosome region was classified as euploid (<30% aneuploidy) or aneuploid (≥30% aneuploidy). Euploid embryos were those showing euploid values for all chromosomes.

### 2.5. Time-Lapse Analysis and Recording of Kinetic and Morphological Parameters

Time-lapse images acquired during the culture period to the point of the biopsy were used for the assessment of the embryo’s development. The time of insemination by ICSI or the hour when the conventional IVF was performed was programmed into the Geri System when the slide was loaded. Embryologists studying time-lapse images annotated the precise timing of the developmental events observed using the viewer. All times were recorded in hours and normalized to the pronuclei fading time (using t_PNf_ as t_0_) to enable us to study the embryos from IVF and ICSI together. Appendix A shows definitions of the dynamic events studied. Morphological features were also tracked, including the morphology of the cells, fragmentation, multinucleation at the 2- or 4-cell stage, pronuclei morphology, direct and reverse cleavage, and the compaction process. Strict guidelines were established for the annotation of video footage. Embryo grading was performed following the recommendations of Alikani et al. [20], the Gardner classification [4], and ESHRE and ASRM guidance [8,21].

A novel parameter is described and studied (start of t2, st_2_), corresponding to the first cytoplasmic movements prior to the first cytokinesis (Appendix A). The cytoplasmic movements were divided into 3 groups depending on their observed phenotype: Pattern 0 for no detectable movements at the microscope level (Appendix A); Pattern 1 for non-patterned movements (Appendix A); and Pattern 2 for circular movements (Appendix A). The initial detection of all these movements in groups 1 and 2 were labeled as st_2_ for each embryo. Group 0 embryos lack st_2_ annotation. Thus, st_2_ annotation corresponds to the first detectable frame of the following movements: halo disappearance, cytoplasmic waves (which usually lead to anarchic blebbing), or circling cytoplasm movements. Specific membrane movements, such as pseudo-furrows or rubbing movements, among others, were not considered, as st_2_ refers only to cytoplasmic movements. All annotations were blinded to ploidy.

### 2.6. Statistical Analysis

Continuous data are expressed as means with standard deviations, and categorical data are expressed as percentages. The relationship between the various parameters studied and the chromosomal dotation was assessed. Categorical data and proportions were analyzed using Chi-square tests. The normality of continuous data was assessed, and a T-test was used for continuous variables. Univariate and multivariate logistic regression analyses were used to create a model for enhanced selection based on aneuploidy, and the odds ratio (OR) was expressed in terms of the 95% confidence interval (95% CI) and significance. Receiver operating characteristic (ROC) curves were used to test the predictive value of all variables included in the model with respect to chromosomal normality. Statistical analysis was performed using GraphPad software. *p* < 0.05 was considered statistically significant.

## 3. Results

In this study, 374 embryos from 85 IVF/ICSI cycles were studied for aneuploidy. The mean age of the female partners was 38.0 years, ranging from 22 to 45 years. The patient demographics, indications for performing genetic testing on the embryos, and cycle characteristics are summarized in Appendix A. A total of 158 embryos were diagnosed as euploid, resulting in a 42.2% global euploidy rate (38.3% per cycle). No differences were observed in this rate when comparing embryos from different fertilization techniques (conventional IVF (38.1%) or ICSI (47.1%)). 

Several morphokinetic parameters showed a high association with ploidy status (Table 1). Euploid embryos appear to be faster in their development, reaching specific division times (st_2_, t_3_, t_5_) and blastocyst formation moments (t_SB_, t_B_) earlier than the aneuploid group. Similarly, some intervals also appear to be shorter in the euploid group (cc3, t_5_–t_2_). The mean times for these parameters for euploid and aneuploidy embryos, respectively, are as follows: st_2_: 1.5 ± 0.9 vs. 1.6 ± 0.7; t_3_: 12.9 ± 3.4 vs. 13.6 ± 2.6; t_5_: 25.5 ± 6.1 vs. 27.1 ± 4.5; t_SB_: 73.8 ± 7.0 vs. 76.3 ± 7.5; t_B_: 83.6 ± 7.4 vs. 86.2 ± 7.6; cc3: 12.5 ± 4.8 vs. 13.6 ± 2.9; t_5_-t_2_: 22.9 ± 6.2 vs. 24.4 ± 4.7. Data from the euploid group are distributed more equally, while the aneuploid population tends to be more dispersed, especially above the median. 

St_2,_ a novel and poorly studied parameter, was shown to be highly discrepant between euploid and aneuploid blastocysts. St_2_ was considered to be the initial cytoplasmatic movement that embryos present prior to the first cell cleavage. These phenomena presented different expressions, divided into three corresponding groups. Some of the embryos (Pattern 0) did not present any detectable movements when observed under the microscope. These embryos did not present halo dissolution or redistribution of the cytoplasmic material, either towards the cortex (as commonly described) or in any other textural change. A small group of embryos lacking a flare or halo, in addition to not showing any other cytoplasmic reorganization, are included in this group. The cleavage to the two-cell stage was smooth (Appendix A). Pattern 1 includes embryos with cytoplasmic or peripheral (adjacent to the membrane) vibration-like movements prior to the first cell division. These movements do not follow any identifiable pattern and have a random appearance. Movements annotated in this group include halo disappearance (Appendix A), cytoplasmic polarization, or random waves (Appendix A), which usually lead to anarchic blebbing. However, specific membrane movements, such as pseudo-furrows or rubbing movements, among others, were not considered; we only focused on cytoplasmic movements. Commonly, these start after the PN breakdown, but we also detected them immediately prior to the pronuclei disappearance (referring specifically to halo fading). When this was the case, st_2_ was annotated and, accordingly, adopted negative values after the PN fading correction was applied. Pattern 2 embryos present clear circular wave movements at the cytoplasm (Appendix A). Circular movements are often repeated, and most of the embryos presenting this cortical rotation exhibit at least 2–3 complete rotations before cell cleavage. The occurrence of each pattern is shown in Table 2 and shows a clear correlation with chromosomal dotation (*p* value < 0.0001). Pattern 2 was highly associated with euploid embryos: 90% of the embryos with circular movements were diagnosed as euploid. In contrast, Pattern 0 was more frequently observed in aneuploid embryos compared to euploid blastocysts (66.3% vs. 33.7%, respectively).

Regarding the morphological variables visible with TL technology, we studied multinucleation, direct or reverse cleavage, the size and symmetry of pronuclei, the presence of exclusion cells, and the compaction process. None of these parameters appear to be significantly associated with the ploidy status.

The correlation between the various morphokinetic timings and intervals studied is represented in a correlation matrix. For euploid embryos, especially in the case of intervals (Figure 1), we found a positive correlation for most of them. Initial divisions, as well as the blastulation process, appear to be significantly involved in chromosomal dotation, with early cleavages presenting a clear correlation with blastulation time. On the contrary, aneuploid embryos showed null or insignificant correlations, with r values that were not statistically different from 0.

Using univariate logistic regression models, we confirmed the parameters that most strongly affect the ploidy status (Table 3). We filtered the most relevant parameters according to all of our results, controlled the data for egg age (as it is a known confounder), and performed a multivariate logistic regression study to develop a model using the Odds ratios (Table 4). The area under the ROC curve was AUC = 0.69, with a 95% confidence interval (0.62 to 0.76), which suggests an existing but moderate predictive ability. Using this system, we were able to identify 49.5% of euploid embryos. However, 80.6% of aneuploid embryos were correctly classified. When looking at the specific weight of each parameter on the model, only the st_2_ pattern significantly affected the outcome as an independent predictive factor. 

## 4. Discussion

This study retrospectively analyzed the relationship between embryo development data and chromosomal dotation, and the results show that morphokinetics of embryo development are related to embryo ploidy. Embryonic aneuploidies may affect the development and cleavage behavior of embryos. By using time-lapse culture systems, we were able to monitor and identify those kinetic events and transient morphological attributes that can help us better select embryos for transfer by predicting embryo ploidy without using PGT. While an improvement in clinical outcomes has been seen over recent years when using TL incubators, meta-analyses comparing these studies show little to no evidence of the improvement of embryo selection when using TL embryo selection software compared to conventional assessments [22]. Therefore, it is necessary to refine TL embryo evaluation systems.

We found no differences between embryos from conventional IVF versus ICSI, either in fertilization or in euploidy rates. The majority of published studies focus on ICSI cycles. We believe that neglecting conventional IVF in these studies is a mistake, as it is an optimal technique for many couples, especially in cases with non-male-factor infertility. Additionally, while ICSI used to be mandatory before PGT studies to avoid sperm DNA contamination, by switching from a D3 to a TE biopsy, this potential risk can be overlooked [23]. By including IVF cycles in our study, we are able to apply the results obtained to all our patients, regardless of the fertilization technique that is suitable for them. We did find differences in maternal age between the euploid and aneuploid groups. While oocyte age is one of the main known factors affecting euploidy, we applied corrections to our prediction models to discard age as a confounding factor.

It is worth mentioning that PGT-A does not increase cumulative pregnancy rates. The probability of achieving a successful pregnancy after transferring a euploid embryo is significantly higher compared to the results of transferring an embryo without genetic analysis. However, it does not have superior outcomes for a whole embryo pool. Focusing on individual embryo transfers and not on cumulative outcomes falsely suggests that PGT-A improves outcomes, but it actually shortens the time and transferences required to achieve these results [24,25]. IVF treatments can be onerous and tedious for patients with fertility problems; they are associated with stress, and they represent a significant economic burden for the patients. Reducing the time and attempts needed for the whole process is a benefit in and of itself, so the advantages of PGT-A for specific patients are undeniable. However, questions continue to be raised regarding the invasiveness of the technique and the question of whether only a specific subset of patients truly benefit more from its usage (e.g., in cases with advanced maternal age or previous miscarriages).

Novel techniques for identifying euploid embryos without the technical and economic drawbacks of invasive and non-invasive preimplantation screening must be found. Reducing transfer failures could lead to fewer patients discontinuing treatment. Nowadays, new approaches for diagnosing embryo ploidy using novel, less invasive techniques are being developed, such as non-invasive preimplantation genetic testing, which is based on the analysis of DNA found within the blastocoel fluid of blastocysts and in spent media culture (niPGT) [26,27,28]. However, despite the significant potential of niPGT, we are a long way from making those procedures accessible to all patients, both because problems related to complexity, technical complications, and limitations remain unsolved and also because of the high economic cost such procedures can entail despite not being diagnostic methods but giving a “recommendation” result instead. A polar body biopsy is also an alternative. This technique is less invasive than TE biopsy, it does not affect embryo morphokinetics, and it has no negative impact on implantation events. However, only maternal genetic information is obtained. Although 90% of human aneuploidies at birth are of maternal origin, the possibility of paternal aneuploidies cannot be dismissed [29].

Meanwhile, the use of algorithms and grading systems based on morphokinetic parameters to predict ploidy can be of great help when selecting embryos for transfer. Even for those patients who do not have a medical recommendation to perform PGT-A, we could increase the success rates of IVF treatments by increasing the probability of transferring euploid embryos. Our results, in line with previously published studies [10,30], show that high-quality embryos also present aneuploidies at a considerable rate (50.3% of AA blastocysts are aneuploidy in our study). Thus, we need more than a static grading system based on quality, from which we can predict the genetic content of the embryo. AI prediction models are, in this sense, promising. However, they are still in the development phases and do not reach the same level of specificity as PGT.

Regarding the morphological markers observed thanks to TL, including pronuclei morphology, multinucleation, chaotic cleavages, or the compaction process, none of the studied variables appeared to be related to the embryo’s ploidy status. These markers were previously identified as critical parameters for blastocyst formation [31,32,33]. However, some seem to also be clearly related to ploidy abnormalities, such as an aberrant number of cell nuclei. Two main hypotheses can explain this fact. First, defective markers are highly associated with bad blastulation, and we only analyze high-quality blastocysts for the PGT study. Those embryos presenting altered parameters affecting blastocyst formation would not reach the optimal stage to be biopsied and, thus, would not have been included in this study. During preimplantation development, there is a progressive loss of abnormal embryos caused by growth arrest or degeneration. Accordingly, the sample size for these variables was extremely small in most of the cases. Second, embryos may present “self-correction” systems for some of these parameters. For example, while excluded cells may be associated with chromosomal defects, this may be the reason that these particular cells are excluded from the blastocyst [34]. It has also been noted that multinucleated cells have the potential for self-correction during the first cleavages and develop euploid blastocysts [35]. Similarly, some studies report that around 20% of trisomies in cleavage-stage embryos are self-corrected and lead to euploid blastocysts [36]. However, in all of these self-correcting cases, there can exist collateral abnormalities such as uniparental disomies.

Focusing on kinetic markers, we found seven timings to be particularly strongly associated with chromosomal status: st_2_, t_3_, t_5_, t_SB_, t_B_, cc3, and t_5_–t_2_. These key markers demonstrate that there are critical steps throughout the whole embryo development process. Our results agree with those of previous studies, which described synchronized and early cleavages, as well as early blastulation, as being valuable for ploidy prognosis [17,18,19]. Furthermore, we performed a test to identify outliers prior to the study. Interestingly, more extreme values were found in the aneuploid population. Thus, abnormal or aberrant behaviors are often related to altered chromosomal content. In most of the studies published, slower blastocyst formation is associated with poorer embryo viability [37]. We also obtained similar results, whereby delayed timings are associated with aberrant embryos. The sooner and shorter those changes are, the higher the probability of euploidy. This delay in development could be explained by the abnormal activation of the spindle assembly checkpoint [38] in those cells presenting aneuploidy, as the aneuploid chromosome encounters difficulties in aligning with the metaphase plate. Additionally, a defective or lax checkpoint during embryonic development may lead to further development with chaotic or aberrant divisions [39].

Curiously, we found higher aneuploidy rates to be associated with delayed blastulation, but we found no difference in the ploidy status between D5 and D6 biopsied embryos. A consistent conclusion has not yet been reached in this ongoing debate. Some authors suggest better clinical outcomes for D5 embryos [40,41], whereas other studies find no differences associated with the embryo transfer day [42,43,44]. A common explanation for lower pregnancy outcomes for D6 transfer is the discordance between the patient endometrium and the embryo stage. Particularly in fresh cycles, when the implantation window is advanced after ovarian stimulation, we face an embryo–endometrium desynchronization scenario, especially in D6 transfers. When only high-quality embryos in fresh transfers were considered, no differences were found between D5 and D6 blastocysts [45]. Low-quality embryos could present with slower development and also be associated with chromosomal anomalies that could impair their viability and the implantation potential of those fresh D6 blastocysts. Again, only high-quality embryos are candidates for embryo biopsy and PGT-A in our clinic, excluding those potentially altered embryos. Moreover, we normalize our study times to the disappearance of PN, suggesting that the impact of D5 vs. D6 potential may not rely on pure timings themselves but rather on the way they progress from one stage to another (especially because we include conventional IVF cycles). Delayed fertilization followed by a chronological and punctual division can result in a euploid and viable D6 blastocyst, while early fertilization followed by decompensated or delayed division steps can be associated with a D5 embryo with a poor prognosis. Thus, delayed blastulation refers to the embryonic inner time spent until the start of or complete blastulation, but it does not directly correlate to “our” D5 or D6.

We found a poorly studied parameter that is of special importance in this context: st_2_. St_2_ refers to the initial cytoplasmic macroscopic observable movements that precede the first cytokinesis. We demonstrated that the start of the t_2_ division is key to the ploidy outcome, as delayed times for this first movement are associated with a higher likelihood of aneuploidy and affect the following cleavages. There are molecular steps required for cell division (maturation, chromosome division, protein machinery, etc.) that can translate into these ooplasmic movements. This organelle reorganization, which prepares the cell to divide, could be more important than the time spent in proper cytokinesis itself. For example, halo disappearance is related to the microtubule-mediated withdrawal of mitochondria and other cytoplasmic components of perinuclear regions, and mitochondrial activity is highly involved in cell cycle regulation [46]. Fertilization cytoplasmic dynamics have been described in detail in relation to the PN appearance, alignment, and fading events and can predict clinical outcomes [47]. Fast cleavage is related to good blastocyst formation with regard to membrane ruffling [48]. St_2_, meanwhile, is an ooplasmic event prior to the first cell cleavage that could also be used in embryo evaluation, as we found it to be related to ploidy. Based on our results, embryos presenting early st_2_ movements are more likely to present euploid complements, while membrane blebbing is usually observed later. Additionally, the nature of the cortex dynamics is important, as we describe three types of movement. A lack of movement is associated with poor blastulation prognosis [49], and we see a tendency towards aneuploidy. Embryos diagnosed as aneuploid in Pattern 0 are almost twice as numerous as the euploid group. While cytoplasmic movements are found in both euploid and aneuploid populations, circular waves, in particular, appear to be a characteristic of euploid embryos. All of the embryos presenting this rotation, except two, were diagnosed as euploid. However, while the statistical power of these results is strong, the proportion of Pattern 2 embryos is small compared to the total sample of embryos studied. A bigger study could be valuable in further corroborating these results.

We also found an interesting correlation between most of the kinetic markers exclusively in euploid embryos. This confirms that aneuploid embryos present altered cleavage and developmental times, not only promoting delayed blastulation but also de-synchronizing most of the sequential steps and division without a clear pattern. This chaotic behavior in aneuploid embryos is reinforced by the conventional grading system, as these embryos present oscillating gradings throughout their development. Our correlation matrix reveals that synchronic even cleavages (which lead to even numbers of cells, e.g., t_4_–t_3_) positively condition blastulation time. In contrast, synchronized and sequential odd cleavages (which lead to odd numbers of cells, e.g., t_3_–t_2_) inversely affect blastocyst formation and are a good prognostic for normal chromosomal content. The logistic regression model derived from this study presents an AUC = 0.69 and can help us to discard aneuploid embryos in 80.6% of cases. This can be useful when trying to select the best embryo for transfer, by rejecting those with a higher probability of aneuploidy. As mentioned above, machine learning algorithms and AI are promising tools, but they are still not fully reliable for ploidy selection. In this study, we tested our embryo populations retrospectively using AI to obtain a score and ranking of ploidy prediction. The present embryo evaluation system based on morphokinetic parameters, including st_2,_ obtains slightly better results than the use of AI-integrated predictors (AUC 0.69 vs. 0.64, respectively) [50]. While it was a small study, together, these results suggest the potential benefit of adding the novel markers described here to embryo selection systems.

Several algorithms and models have been developed to predict embryo ploidy [18,51]. However, some studies found no correlation in distinguishing euploidy [52]. Not only that, but some authors applied published models and failed to obtain accurate predictions for their cohort of blastocysts [53]. Considering all these discrepancies, it is imperative that all TL embryologists are meticulously trained to identify all of the developmental markers equally. Not only may the annotation vary slightly from one center to another (where slight variation is enough to invalidate a given model), but external factors may also bias the developmental morphokinetics (e.g., ovarian stimulation, temperature, pH, or culture media, among others). Therefore, we contend that it is useful not to apply an external model but to identify key markers that may condition every cohort independently of the center and laboratory. When using AI, the decision-making process of machine learning models is a black box. We cannot overlook the fact that AI could be taking into account parameters still unknown to embryologists, such as st_2_. Thus, it is crucial that we decipher these parameters and optimize evaluation systems, both to improve embryologists’ work and to optimize future artificial systems.

While none of these promising methods are fully developed, we can (or must) improve embryologists’ grading tools for selecting the best embryos for transfer. Embryologists remain essential for embryo selection. While it could be helpful to have software helping them to decide, it is crucial that we understand what is behind these decisions. Identifying novel developmental events can help us to better understand AI models and ploidy implications. We present st_2_ as an improvement tool to include in evaluation systems, paying special attention to the movement type (circular, not patterned or absent). This study has some limitations due to the sample size in some variables or groups and the retrospective nature of the analysis. Future studies, including large prospective RCTs, could be useful in maximizing time-lapse technology embryo assessments as a non-invasive, cost-effective alternative to PGT-A.

## 5. Conclusions

Morphokinetic analysis is not able to detect aneuploid embryos as accurately as PGT does, but it has the potential to identify the most suitable embryos and reduce the time to pregnancy. We have found two potential tools that can help to select the best embryo for transfer. (i) Our study demonstrates the importance of the relation between a wider group of developmental stages instead of focusing on one or two markers. By paying close attention to the correlation between crucial markers, especially the synchronicity and sequentially of the cleavages, we can discard potentially aneuploid embryos. (ii) We describe st_2_ as a novel parameter that strongly influences the developmental progress of the following steps and which is strongly associated with chromosomal status. Specific cytoplasmic waving patterns correlate with ploidy diagnosis, with circling movements being the most strongly correlated with euploidy. Including this marker in conventional grading systems, as well as in selection models and algorithms, will lead to better results by improving embryo selection.

## Figures and Tables

**Figure 1 jcm-12-02983-f001:**
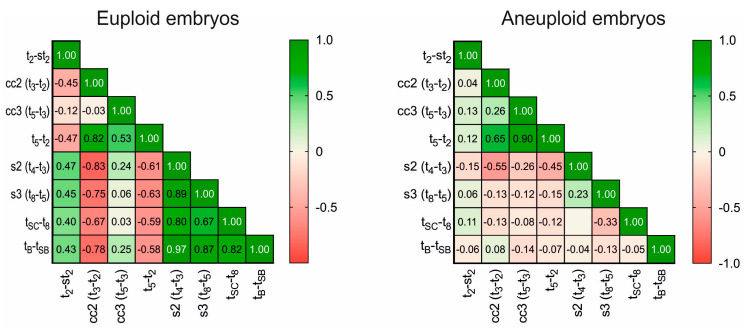
Correlation matrix of the studied interval parameters. The left panel shows the matrix obtained for euploid embryos; the right shows that obtained for aneuploid embryos. Numbers inside cells show r values. The color legend shows the degree of correlation: green (1.0), positive correlation; red (−1.0), negative correlation; 0, no correlation.

**Table 1 jcm-12-02983-t001:** Annotations for morphokinetic parameters were analyzed using time-lapse technology.

	Euploid Embryos		Aneuploid Embryos		*p*-Value
	Mean (h) ± 95% CI	*n*	Mean (h) ± 95% CI	*n*	
**st_2_**	1.5 ± 0.9	128	1.6 ± 0.7	157	**0.03**
t_2_	2.6 ± 0.5	156	2.7 ± 0.6	216	ns
**t_3_**	12.9 ± 3.4	157	13.6 ± 2.6	215	**0.05**
t_4_	14.4 ± 2.4	155	14.6 ± 2.2	213	ns
**t_5_**	25.5 ± 6.1	156	27.1 ± 4.5	215	**0.004**
t_8_	34.8 ± 7.4	154	35.2 ± 7.3	213	ns
t_SC_	54.2 ± 11.5	156	54.7 ± 11.3	216	ns
**t_SB_**	73.8 ± 7.0	158	76.3 ± 7.5	216	**0.001**
**t_B_**	83.6 ± 7.4	151	86.2 ± 7.6	212	**0.001**
t_2_–st_2_	1.2 ± 0.9	128	1.1 ± 0.8	157	ns
cc2 (t_3_–t_2_)	10.4 ± 3.4	156	10.8 ± 2.9	215	ns
**cc3 (t_5_–t_3_)**	12.5 ± 4.8	156	13.6 ± 2.9	215	**0.006**
**t_5_–t_2_**	22.9 ± 6.2	156	24.4 ± 4.7	215	**0.008**
s2 (t_4_–t_3_)	1.5 ± 1.0	155	1.2 ± 1.3	213	ns
s3 (t_8_–t_5_)	9.2 ± 3.7	154	8.3 ± 3.6	213	ns
t_SC_–t_8_	19.6 ± 5.5	154	19.5 ± 5.8	213	ns
t_B_–t_SB_	9.9 ± 2.6	151	10.0 ± 2.9	212	ns

ns: not significant.

**Table 2 jcm-12-02983-t002:** Occurrence of the different cytoplasmic patterns of st_2_ in euploid and aneuploid embryos.

	Euploid Embryos	Aneuploid Embryos	*p*-Value
Pattern 0	30 (33.7%)	59 (66.3%)	<0.0001
Pattern 1	110 (41.5%)	155 (58.5%)	
Pattern 2	18 (90.0%)	2 (10.0%)	

**Table 3 jcm-12-02983-t003:** Univariate logistic regression analysis.

Parameter	Odds Ratio	95% IC	*p*-Value
**st_2_**	0.763	1.017–1.936	**0.04**
t_2_	1.265	0.887–1.844	ns
t_3_	1.061	0.990–1.138	ns
t_4_	1.014	0.926–1.110	ns
**t_5_**	1.058	0.017–0.097	**0.005**
t_8_	1.008	0.980–10.37	ns
t_SC_	1.005	0.988–1.023	ns
**t_SB_**	1.051	1.021–1.083	**<0.001**
**t_B_**	1.049	1.020–1.081	**<0.001**
**t_2_–st_2_**	0.715	0.537–0.899	**<0.0001**
cc2 (t_3_–t_2_)	1.043	0.976–1.115	ns
**cc3 (t_5_–t_3_)**	1.080	1.022–1.146	**0.006**
**t_5_–t_2_**	1.053	1.013–1.096	**0.008**
s2 (t_4_–t_3_)	0.931	0.845–1.023	ns
s3 (t_8_–t_5_)	0.983	0.955–1.013	ns
t_SC_–t_8_	1.000	0.983–1.017	ns
t_B_–t_SB_	1.006	0.961–1.056	ns
**st_2_ pattern**	0.635	0.468–0.845	**0.001**

ns: not significant.

**Table 4 jcm-12-02983-t004:** Multivariate logistic regression analysis.

Parameter	Odds Ratio	95% IC	*p*–Value
st_2_	0.979	0.642–1.493	ns
t_5_	0.655	0.362–1.143	ns
t_SB_	0.968	0.899–1.044	ns
t_B_	0.984	0.916–1.052	ns
cc3 (t_5_–t_3_)	1.380	0.820–2.398	ns
(t_5_–t_2_)	1.012	0.8725–1.167	ns
**st_2_ pattern**	1.648	1.088–2.539	**0.02**

ns: not significant.

## Data Availability

Not applicable.

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
