# Peer review of "Novel Time-Lapse Parameters Correlate with Embryo Ploidy and Suggest an Improvement in Non-Invasive Embryo Selection"

_jcm, 2023, doi:10.3390/jcm12082983_

Round 1

Reviewer 1 Report

This study aimed to evaluate the correlations between time-lapse parameters and ploidy status. The main results demonstrated a novel kinetic parameter, i.e., st2, was a critical factor associated with embryo ploidy. The ROC curve analysis revealed the kinetic parameters might be useful for ploidy prediction (AUC = 0.69). Some questions in the following paragraph need further explanation or clarification.

1.   Experimental designs:

a.   There were different platforms applied for PGT-A in this study, and only euploid and aneuploid blastocysts were included for analysis. The authors need to clearly define the “euploidy” and “aneuploidy”. Were the mosaic embryos excluded/included in this study?

b.   Could the parameter st2 be objectively determined? The authors should have more specific and detailed description for this parameter. It seems to be similar to the timings of starting or finishing granule redistribution after cytoplasmic halo formation. The reviewer would suggest the authors to do the advanced analysis about halo characteristics (kinetics or morphological features) and discuss its correlations with ploidy status using TL monitoring.

2.   Materials and methods:

a.   Line 143: “normalized by the pronuclei fading time to study together embryos from IVF and ICSI.” Did the authors use tPNf as t0? Please clarify it.

b.   Line 169: “t-test was used for continuous variables.” Please provide the information about normality assumption of the collected data or use nonparametric statistics.

c.   Line 170: “logistic regression analysis were applied to create a model for enhanced selection…” Multiple blastocysts might come from the same couples. The authors should consider to use the suitable analysis method, such as generalized estimating equation, for repeated measures.

3.   Main results and tables:

a.   Table 1:

               i.    Did the blastocysts with pattern 0 of cytoplasmic movement were involved in the Table 1? Please provide the sample numbers for the Tables.

             ii.    According to our experience, cytoplasmic movement should be found before PN fading. Therefore, most or part of st2 might be less than 0 after adjustment of tPNf. Did the authors just observe the occurrence of cytoplasmic movement after PN fading?   

b.   Table 3:

               i.    The values in the table (e.g., st2) are not correct.

             ii.    Several parameters were significantly different between euploid and aneuploid blastocysts in Table 1, but shown to be not associated with ploidy status in Table 3, such as t3, t5, tSB, and cc3. Please have the explanation for these results.

           iii.    st2 patterns should be analyzed as categorical variables.

c.   Lines 248-250: “We filtered the most relevant parameters according to all of our results, controlled data for egg age (as it is a known confounder) …(Table 4).” Did the authors evaluate the correlations of individual kinetic parameters with euploidy status by adjustment of patient age? In the retrospective study, more patient or cycle confounders related embryonic ploidy should be considered, such as COH methods, male infertility factors, male age, indication of infertility, serum hormones, BMI and so on.

d.   Lines 251-252: “The area under the ROC curve was AUC = 0.69, with 95% confidence interval (0.62 to 0.76)…” It is uncertain which parameters are used to predict ploidy. What are the cut-off values for the variables to deselect aneuploid embryos?

Author Response

Response to Reviewer 1

Thank you very much for your revision work and important comments. We modified our work in accordance with your suggestions. We have also proof-read the manuscript to improve the possible English language and style fine/minor typos in the text, as suggested.

Comments and Suggestions for Authors (Reviewer 1)

This study aimed to evaluate the correlations between time-lapse parameters and ploidy status. The main results demonstrated a novel kinetic parameter, i.e., st2, was a critical factor associated with embryo ploidy. The ROC curve analysis revealed the kinetic parameters might be useful for ploidy prediction (AUC = 0.69). Some questions in the following paragraph need further explanation or clarification.

  1. Experimental designs:
  2. There were different platforms applied for PGT-A in this study, and only euploid and aneuploid blastocysts were included for analysis. The authors need to clearly define the “euploidy” and “aneuploidy”. Were the mosaic embryos excluded/included in this study?

Thank you very much for your advice. We have clarified the inclusion/exclusion criteria for mosaic embryos in the material and methods, as well as the euploidy and aneuploidy definitions (lines 142-148, in red). Embryos were classified by the genetic analysis company following this classification criteria: embryos with copy-number deviations below 30% were classified as euploid; embryos with copy-number deviations above 30% were classified as aneuploid. Mosaic embryos (nowadays considered as embryos showing copy-number deviations between 30-70%) were reported to us as aneuploid, since these embryos correspond to cycles that took place between 2018 and 2020 and mosaic embryos were not specifically reported to us at that moment.

  1. Could the parameter st2 be objectively determined? The authors should have more specific and detailed description for this parameter. It seems to be similar to the timings of starting or finishing granule redistribution after cytoplasmic halo formation. The reviewer would suggest the authors to do the advanced analysis about halo characteristics (kinetics or morphological features) and discuss its correlations with ploidy status using TL monitoring.

Thank you very much for your advice. We have clarified the definition of st2 in material and methods (lines 169-173, in red), based on your suggestion.

These movements are indeed similar to the redistribution of cytoplasmic halos in some cases, as we suggest in the discussion, but also include different cytoplasmic reorganizations. Although there are several studies published on halo formation and kinetics, we found no previous literature describing or relating them to euploidy. Moreover, the pattern classification of both halo disappearance but also cytoplasmic movements described here is novel. We highlight in this study the importance of these movements (both time and pattern). We are now designing future studies on specific kinetics or morphology of these st2 movements, but will require the help of advanced image processor or artificial intelligence software to be able to describe these movements in detail. We consider this “preliminary” results important per se, as the parameters described here are not being considered in embryo evaluation at the moment and they can improve embryo selection (e.g. circular movements show a clear correlation with euploidy)

  1. Materials and methods:
  2. Line 143: “normalized by the pronuclei fading time to study together embryos from IVF and ICSI.” Did the authors use tPNf as t0? Please clarify it.

Thank you. Done (line 155, in red)

  1. Line 169: “t-test was used for continuous variables.” Please provide the information about normality assumption of the collected data or use nonparametric statistics.

Thank you. Done (line 187, in red)

  1. Line 170: “logistic regression analysis were applied to create a model for enhanced selection…” Multiple blastocysts might come from the same couples. The authors should consider to use the suitable analysis method, such as generalized estimating equation, for repeated measures.

Thank you for your advice. The reviewer is correct that multiple blastocysts might come from the same couple, and therefore we cannot discard that this common origin might influence somehow the embryo development of that specific cohort. However, each embryo has its own chromosomal dotation, and one couple can have both euploid and aneuploid embryos that we must study separately, as it is the outcome of our study. Therefore, our statisticians team believe they should not be considered repeated measures of the same patient. For this reason, and due to the binary outcome variable and combined categorical and continuous independent variables, our statisticians believe logistic regression analysis is an appropriate method to analyze our data.

  1. Main results and tables:
  2. Table 1:
  3. Did the blastocysts with pattern 0 of cytoplasmic movement were involved in the Table 1?

Blastocysts with pattern 0 were included in Table 1, but they do not have st2 annotated (as they don’t present any st2 movements). Thus, the N of st2 and t2-st2 variables is always smaller than the rest of variables.

Please provide the sample numbers for the Tables.

We are sorry, we do not understand what the referee refers to by this suggestion. The sample size of each variable and group is already shown in all tables (in Table 1 they are provided in the N column, highlighted now in red).

  1. According to our experience, cytoplasmic movement should be found before PN fading. Therefore, most or part of st2 might be less than 0 after adjustment of tPNf. Did the authors just observe the occurrence of cytoplasmic movement after PN fading?

Thank you, we have modified the text to better clarify this issue (line 239-241 in red).

  1. Table 3:
  2. The values in the table (e.g., st2) are not correct.

Thank you very much. We have modified Table 3 to correct the format mistake that was affecting the rows.

  1. Several parameters were significantly different between euploid and aneuploid blastocysts in Table 1, but shown to be not associated with ploidy status in Table 3, such as t3, t5, tSB, and cc3. Please have the explanation for these results.

Thank you very much. Most of these disassociations were due to the format error of table 3. Now, with these being solved, only t3 appears as significantly different between euploid and aneuploid blastocysts in Table 1, but not in Table 3. As t3 was the least significant variable in table 1 (p value 0.05), it makes sense for it to be the least explanatory variable when looking at their relationship with the ploidy outcome.

   iii.    st2 patterns should be analyzed as categorical variables.

Thank you for your advice. St2 patterns are, indeed, analyzed as categorical variables (both in the occurrence of Table 2 and in the logistic regression of Table 3).

  1. Lines 248-250: “We filtered the most relevant parameters according to all of our results, controlled data for egg age (as it is a known confounder) …(Table 4).” Did the authors evaluate the correlations of individual kinetic parameters with euploidy status by adjustment of patient age? In the retrospective study, more patient or cycle confounders related embryonic ploidy should be considered, such as COH methods, male infertility factors, male age, indication of infertility, serum hormones, BMI and so on.

Thank you very much for your suggestion. The reviewer is correct, there are other known confounders that could be related to the ploidy outcome. However, we are not studying the origin of the aneuploidies found, but trying to identify morphokinetic parameters that can help embryologist select the best embryo for transfer. Therefore, our study focusses on recognizing and describing these parameters, independently of what is causing the aneuploidy behind them.

Similarly, age is known to affect ploidy and, accordingly, our euploid and aneuploid groups present different mean female age. However, we want to be able to identify aneuploid embryos, disregarding their origin. This is the reason we did not adjust individual kinetic parameters for age or any other known confounding factor, as we want to understand what does aneuploidy cause on the embryo dynamics, regardless what the aneuploidy is caused by. We did, however, include female age in the multivariate logistic regression model, and still found st2 pattern as highly involved, suggesting a high implication.   

  1. Lines 251-252: “The area under the ROC curve was AUC = 0.69, with 95% confidence interval (0.62 to 0.76)…” It is uncertain which parameters are used to predict ploidy. What are the cut-off values for the variables to deselect aneuploid embryos?

Thank you for your comment. The reviewer is correct, it is uncertain which parameters are used to predict ploidy as we do not present a prediction model with a specific formula with weights/coefficient for each variable. But with using the multivariate logistic regression, we find the most implicated variables (in this case, st2 pattern). The aim of this study is not to present a model and we do not present specific cut-off values, as we believe inter-clinician and especially inter-center variability invalidate these methods, as previously proven when applying external models to different clinics (lines 451-453). We propose novel parameters and an improved evaluation approach for daily embryo evaluation performed by embryologist. Instead of using an external model, we suggest including st2 (especially focusing on the pattern) and looking at the correlation and whole development of the embryo, rather than a particular cleavage timing.

Reviewer 2 Report

In this study the authors assessed morphokinetic parameters of 374 blastocysts from preimplantation genetic testing cycles. They describe the new morphokinetic parameter “st2” as highly implicated in ploidy status and cytoplasmic movement patterns associated with ploidy status. The authors conclude that optimizing the indicators to select the most suitable blastocyst could reduce the time to pregnancy and thereby avoiding invasive and expensive methods.

In general, the manuscript is clearly written, methods and results are well described. In my opinion, the study is interesting and straightforward to publish. However, I recommend the following minor changes, that will probably further improve the manuscript:

Introduction

·         Line 43: One word is missing. chromosomal aberrations?

Material & Methods

·         Company names and location should be mentioned for all products used (e.g. FSH, GnRH antagonist, etc.)

·         Line 106: Describe the decision to use hCG versus GnRH agonist for triggering.

·         Line 111: Watch out for subscript and superscript (37oC, CO2, O2)

Results:

·         Line 178: IVF/ICSI cycles

·         Table 2: p values for patterns 1 & 2 are missing.

Discussion:

·         A more critical insight in TL is needed (e.g. Armstrong et al. 2015, doi: 10.1002/14651858.CD011320.pub2)

·         Information on alternatives to invasive methods like polar body biopsy should be mentioned (doi: 10.1007/s10815-018-1207-4)

·         Strengths and limitations are missing.

General comments:

·         A few typos exit within the manuscript. I would highly recommend having the manuscript proof-read once again.

Author Response

Response to Reviewer 2

Thank you very much for your revision work and important comments. We modified our work in accordance with your suggestions.

Comments and Suggestions for Authors

In this study the authors assessed morphokinetic parameters of 374 blastocysts from preimplantation genetic testing cycles. They describe the new morphokinetic parameter “st2” as highly implicated in ploidy status and cytoplasmic movement patterns associated with ploidy status. The authors conclude that optimizing the indicators to select the most suitable blastocyst could reduce the time to pregnancy and thereby avoiding invasive and expensive methods.

In general, the manuscript is clearly written, methods and results are well described. In my opinion, the study is interesting and straightforward to publish. However, I recommend the following minor changes, that will probably further improve the manuscript:

Introduction

  • Line 43: One word is missing. chromosomal aberrations?

Thank you. Done (line 43, in blue)

Material & Methods

  • Company names and location should be mentioned for all products used (e.g. FSH, GnRH antagonist, etc.)

Thank you. Company names and locations have been added in lines 105-110, in blue.

  • Line 106: Describe the decision to use hCG versus GnRH agonist for triggering.

Thank you. A description for de decision to use hCG versus GnRH has been added (line 110-112, in blue).

  • Line 111: Watch out for subscript and superscript (37oC, CO2, O2)

Thank you. Done.

Results:

  • Line 178: IVF/ICSI cycles

Thank you. Done (line 197, in blue).

  • Table 2: p values for patterns 1 & 2 are missing.

Thank you for your comment. P-value in Table 2 refers to the significance of the Chi-square test among the 3 patterns, showing that there are differences between the three groups. Table column has been combined to clarify it, showing a unique p-value for all the groups.

Discussion:

  • A more critical insight in TL is needed (e.g. Armstrong et al. 2015, doi: 10.1002/14651858.CD011320.pub2)

Thank you for your advice. We have added some sentences in the Discussion (lines 288-292) and cited the suggested reference.

  • Information on alternatives to invasive methods like polar body biopsy should be mentioned (doi: 10.1007/s10815-018-1207-4)

Thank you for your advice. We have added a paragraph in the Discussion (lines 328-333) and cited the suggested reference.

  • Strengths and limitations are missing.

Thank you for your advice. Discussion has been edited to clearly state the strengths and limitations of the study (lines 467-475, in blue).

General comments:

  • A few typos exit within the manuscript. I would highly recommend having the manuscript proof-read once again.

Thank you for your advice. We have proof-read the manuscript with an English editing service to correct the English typos in the text, as suggested.

Round 2

Reviewer 1 Report

The reviewer has no more questions for this study.